# GRGB rPPG: An Efficient Low-Complexity Remote Photoplethysmography-Based Algorithm for Heart Rate Estimation

**DOI:** 10.3390/bioengineering10020243

**Published:** 2023-02-12

**Authors:** Fridolin Haugg, Mohamed Elgendi, Carlo Menon

**Affiliations:** 1Biomedical and Mobile Health Technology Lab, ETH Zurich, 8008 Zurich, Switzerland; 2Department of Mechanical Engineering, Karlsruher Institute for Technology, 76131 Karlsruhe, Germany

**Keywords:** Imaging PPG, imaging photoplethysmogram, camera-based photoplethysmography, remote health monitoring, light interaction with tissue, vasomotor reactivity, vascular regulation, early diagnostic methods, imaging and sensing, optical imaging

## Abstract

Remote photoplethysmography (rPPG) is a promising contactless technology that uses videos of faces to extract health parameters, such as heart rate. Several methods for transforming red, green, and blue (RGB) video signals into rPPG signals have been introduced in the existing literature. The RGB signals represent variations in the reflected luminance from the skin surface of an individual over a given period of time. These methods attempt to find the best combination of color channels to reconstruct an rPPG signal. Usually, rPPG methods use a combination of prepossessed color channels to convert the three RGB signals to one rPPG signal that is most influenced by blood volume changes. This study examined simple yet effective methods to convert the RGB to rPPG, relying only on RGB signals without applying complex mathematical models or machine learning algorithms. A new method, GRGB rPPG, was proposed that outperformed most machine-learning-based rPPG methods and was robust to indoor lighting and participant motion. Moreover, the proposed method estimated the heart rate better than well-established rPPG methods. This paper also discusses the results and provides recommendations for further research.

## 1. Introduction

Variations in blood volume or blood flow occur with every heartbeat [1,2]. Photoplethysmography (PPG) is the measurement of those changes that are related to the pulsating blood volume in the skin tissue [1,2]. Remote photoplethysmography (rPPG) is PPG without skin. The technology is very promising for large populations; rPPG signals can be captured with a simple red–green–blue (RGB) camera, which means that smartphones or laptops can be used to record them. For most people, smartphones, and laptops with integrated cameras are widely available and much more accessible than pulse oximeters or cuff-based blood pressure monitors [3,4]. Several health parameters, such as heart rate (HR), heart rate variability (HRV), and blood pressure (BP), can be determined from a PPG signal [5,6].

As cardiovascular diseases (CVDs) are the leading cause of death worldwide, [7] cardiovascular health parameters can help detect or assess CVDs at an early stage. Thus, large populations could benefit from regular assessment of cardiovascular health parameters if this were done with consumer-grade RGB cameras and potentially become a device outside the clinical environment that could be used as part of the patient’s routine life [8,9]. Moreover, concerning coronavirus disease 2019, rPPG is very promising as it allows contactless measurement of cardiovascular health parameters. Furthermore, patients can perform the measurements from home without any additional risk of infection. To determine cardiovascular health parameters accurately, it is necessary to obtain a high-quality rPPG signal. For these reasons, researchers have recently investigated a variety of rPPG methods.

The reflected light from some regions of the skin of the human face, such as the forehead and cheeks, is affected by blood flow under the skin. These blood flow changes can be recorded with an RGB camera, followed by the transformation of an RGB signal into an rPPG signal. However, there is a disagreement in recent literature about how this is best done. Several approaches have been proposed and compared for different behavioral and environmental conditions. Several studies have compared different rPPG methods using various datasets for HR estimation [10,11,12]. However, it is often not possible to prove why one rPPG method is better than another; only the results can be compared. Researchers have tried approaches, such as different decomposition methods (e.g., principal component analysis (PCA) and independent component analysis (ICA)), transformations in other color models, and varying color channel normalization techniques. These methods perform differently on each applied dataset [10].

Note that the contact-based PPG signal is typically recorded with a pulse oximeter consisting of a light source and a photodetector. An advantage here is that the environmental conditions are negligible due to direct skin contact; therefore, the heart rate is easily detected with high accuracy. On the contrary, the rPPG requires capturing video of the face and processing it to extract the blood flow information and map it to rPPG. Recently, more complex and advanced signal processing techniques are increasingly being developed to obtain high-quality rPPG signals. This adds more complexity to the processing and computational power of the camera-based application. In the present study, we are attempting to produce a new simple but efficient rPPG method based on raw RGB video signals without the need for complex statistical modeling and machine learning algorithms. We then compared the proposed method with well-established methods to examine its performance.

## 2. Methodology

### 2.1. Hypothesis

Based on the literature, the green color channel has been verified to have the highest similarity to a PPG signal in comparison to the red and blue color channels [13]. Furthermore, it is assumed that the light intensity fluctuations caused by skin movements influence all color channels equally. Therefore, this study hypothesized that the ratio of the green-to-red (GR) channel and the ratio of the green-to-blue (GB) channel, or the sum of these ratios (GR + GB), may improve the quality of the constructed rPPG signals.

### 2.2. Dataset

The LGI-PGGI dataset from Pilz et al. [14] was used for the evaluation. It includes videos of the participants’ faces with the referenced fingertip PPG signal. The videos from six participants were publicly accessible, each with four distinct settings, as follows

Resting: indoor participant, sitting, head barely moving.Gym: on a bicycle ergometer, participant exercising indoors.Talk: urban setting with daylight and conversation.Rotation: the participant makes irrational head motions in an indoor setting.

Each participant had one recording of all four video settings. In each video, the participant’s face is at the center of the video. Of the six participants, five were men and one was a woman. The duration was more than 1 min and the resolution was 640×480 for each video. The average sampling rate was 60 Hz for the pulse oximeter and 25 Hz for the RGB camera. Six windows per video were constructed by simply using the first minute of each video.

### 2.3. Pipeline and rPPG Methods

The RGB signal was obtained from the average RGB values of the pixels in the three regions of interest (ROIs). The ROIs were the forehead, left cheek, and right cheek in each frame, which were determined with landmarks from the MediaPipe Face Mesh [15]. The forehead and cheeks were found to be the most promising ROIs for rPPG in two independent ROI studies conducted by Sungjun et al. [16] and Dae-Yeol et al. [17]. The exact landmarks were (107, 66, 69, 109, 10, 338, 299, 296, 336, 9) for the forehead, (118, 119, 100, 126, 209, 49, 129, 203, 205, 50) for the left cheek, and (347, 348, 329, 355, 429, 279, 358, 423, 425, 280) for the right cheek, as shown in Figure 1.

Each video consists of a time series of frames. Each frame is composed of a number of pixels (e.g., full HD: 1920×1080 pixels). The proportions of the three main colors red, green, and blue define a color. This RGB scheme is based on additive color mixing and is only one of the numerous color models available; however, it is very popular, and a large number of video recordings are based on it.

This study aimed to observe the conversion of an RGB signal to an rPPG signal. We did not optimize the entire pipeline for optimal beats per minute (BPM) estimation. A detrend or a Butterworth bandpass filter, for example, can improve the rPPG signal [18,19]. The pipeline consisted of only one filter and did not use other optimization options. Our three proposed rPPG methods are summarized in Table 1. The RGB channels were divided into six 10-s, non-overlaying windows. For each window, a Butterworth bandpass filter of order six, ranging from 0.65 Hz to 4 Hz, was applied.

We compared our rPPG methods with the most recent rPPG methods: GREEN [13], ICA [20], PCA [21], the chrominance-base method (CHROM) [22], blood volume pulse (PBV) [23], plane orthogonal to the skin (POS) [11], local group invariance (LGI) [14], and orthogonal matrix image transformation (OMIT) [24]. All mentioned rPPG methods are implemented in the Python toolbox, pyVHR, from Boccignone et al. [25]. It is important to note that PCA and ICA are blind-source separation-based rPPG algorithms. In this study, the rPPG signal was the second component of ICA and PCA.

The RGB signal then consists of the average value of the ROI, resulting in three time series (red, green, and blue). The time-dependent representation $\mathrm{RGB}\left(t\right)={\left(R\left(t\right),B\left(t\right),G\left(t\right)\right)}^{T}$ and the matrix representation RGB={R,B,G} is used.

The GREEN [13] rPPG method uses the green channel to construct the rPPG signal. Of the three RGB color channels, the green channel has the highest similarity with the PPG signal. This is based on the observation that hemoglobin, a protein involved in the transportation of oxygen, reaches its maximum level of absorption near green light [26]. The resulting rPPG signal is equal to the G color channel
rPPG(t)=G(t).

The RGB signal is subjected to ICA [20] to recover three different source signals. Cardoso [27] invented the joint approximate diagonalization of eigenmatrices (JADE) technique, which was applied to the ICA method. Tensor methods are used in the ICA approach, which involves the joint diagonalization of cumulant matrices and the use of fourth-order cumulant tensors. The solution approximates the statistical independence of the sources (to the fourth order). Although the ICA components are not ordered, the second component usually contains a substantial rPPG signal and is used in the ICA rPPG method. The temporal RGB traces is expressed as follows:
RGB(t)=A·Z(t), where A is a memoryless mixture matrix of the latent sources $Z\left(t\right)={\left(z_{1}\left(t\right),z_{2}\left(t\right),z_{3}\left(t\right)\right)}^{T}$. The source recovery problem can be recast as an estimation of the demixed matrix problem $W=A^{-1}$. This is a problem of blind separation that was solved by Cardoso [27]. The highest correlation with the PPG signal was found in the second component
$$\mathrm{rPPG}\left(t\right)=z_{2}\left(t\right).$$

PCA is a technique that is frequently used for data reduction in pattern recognition and signal processing. PCA can also be used to obtain an rPPG signal from an RGB signal [21]. PCA and ICA try to find the most periodic signals. However, movement or other disturbances can also be periodic.

In PCA, the idea is to find the components $s_{1}\left(t\right),s_{2}\left(t\right),\cdots s_{N}\left(t\right)$ of S(t), with the maximum amount of variance possible by *N* linearly transformed components. The principal components are given by
$$S\left(t\right)=W^{T}\cdot \mathrm{RGB}\left(t\right),$$ where *W* represents the de-mixing matrix which is estimated by PCA, such that the greatest possible variance lies on the first coordinate. For the rPPG signal the second component [21] is often used:
$$\mathrm{rPPG}\left(t\right)=s_{2}\left(t\right).$$

The OMIT method calculates the rPPG signal by generating an orthogonal matrix with linearly uncorrelated components representing the orthogonal components in the RGB signal, relying on matrix decomposition.

The QR factorization [28] method is used to find linear least-squares solutions in the RGB space with Householder Reflections [29]. The resulting equation is:
RGB=Q·T, where *Q* is the orthogonal basis for T(RGB) and *T* is an upper-right triangular–invertible matrix that displays the relationships between the columns in *Q*.

The CHROM [22] method removes the noise by light reflection by color difference channel normalization. The method is based on the idea that the ratio of two normalized color channels would not be affected by the movement of light because different intensities would affect all channels equally. From the RGB signal, the method builds two orthogonal chrominance signals: $X_{CHROM}\left(t\right)=R\left(t\right)-G\left(t\right)$ and $Y_{CHROM}\left(t\right)=0.5R\left(t\right)+0.5G\left(t\right)-B\left(t\right)$. The resulting rPPG signal is given by:
$$\mathrm{rPPG}\left(t\right)=X_{CHROM}\left(t\right)-\alpha Y_{chrom}\left(t\right),$$ where $\alpha =\sigma \left(X_{CHROM}\right)/\sigma \left(Y_{CHROM}\right)$ and σ() is the standard deviation.

The POS method [11] uses the plane orthogonal to the skin tone in the RGB signal to calculate rPPG. The resulting rPPG signal is given by:
$$\mathrm{rPPG}\left(t\right)=X_{POS}\left(t\right)-\alpha Y_{POS}\left(t\right),$$ where $X_{POS}\left(t\right)=R\left(t\right)-B\left(t\right)$, $Y_{POS}\left(t\right)=G\left(t\right)+B\left(t\right)-2R\left(t\right)$, and α is calculated as in CHROM.

To distinguish pulse-induced color changes from motion noise, the PBV [23] approach derives the rPPG signal in the RGB data with blood volume pulse fluctuations. Differences in a normalized RGB space along a very accurate vector result from the different absorption spectra of arterial blood and bloodless skin. The blood volume pulse vector results from the vector representation of the preprocessed color channels.
$$P_{BV}=\frac{\left[\sigma \left(R_{n}\right),\sigma \left(G_{n}\right),\sigma \left(B_{n}\right)\right]}{\sqrt{\sigma^{2}\left(R_{n}\right)+\sigma^{2}\left(G_{n}\right)+\sigma^{2}\left(B_{n}\right)}}.$$

For a specific light spectrum and specific transfer properties of the optical filters in the camera, the $\vec{P_{BV}}$ can be identified. An rPPG algorithm with high motion robustness can be created using this “signature” The rPPG signal is calculated by the projection:rPPG(t)=M·RGB(t), where *M* is the orthogonal matrix:
$$M=kP_{BV}{\left(\mathrm{RGB}\cdot {\mathrm{RGB}}^{T}\right)}^{-1},$$ where *k* is a normalization factor.

The LGI [14] calculates an rPPG signal with a robust algorithm using local transformations. The rPPG signal is computed by:
rPPG(t)=D·RGB(t), where *D* is:
$$D=I-U\cdot U^{T},$$ and where *U* is the result of a single value decomposition of the RGB signal. The identity matrix is *I*.

It can be stated that every established method, besides GREEN [13], requires more calculation steps than the simple ratio-based methods presented in this paper.

### 2.4. Evaluation Metric

To determine the absolute BPM difference (|Δ BPM|), a power spectrum (PS) analysis was conducted. The fast Fourier transform of the autocorrelation function is a definition that is commonly used for a PS [30]. The outcome is shown as a plot of signal power against frequency. Since the frequency of the peak in the PS plot corresponds to the HR, this analysis is often performed for PPG and rPPG signals. The resulting |Δ BPM| is the frequency difference of the peak in the PS between the estimated rPPG signal and the reference fingertip signal from a normalized 10-second window.

## 3. Results

Figure 2 shows an example of the 10-second window of the rPPG signal from the proposed methods. In the figure, the systolic peaks are recognizable. However, it is not easy to visually compare the results; therefore, the power spectrum is plotted on the right side of each signal. The heart rate derived from rPPG constructed using the GR method was close to the heart rate calculated via the fingertip PPG signal, with |ΔBPM|=1.46, while the rPPG constructed using the GB method achieved |ΔBPM|=41.02. Interestingly, the GRGB achieved |ΔBPM|=0.73.

The video setting ’Gym’ is one of the more challenging video settings to determine heart rate. Additional noise sources, compared to the other video settings, are many small movements of the face which come from performing an activity, sweat on the face of the participant, and the heart rate does not have to be constant over a 10-second window. These noise sources can be seen in Figure 2, the heart rate peak of the rPPG signal, which does not stand out from the surrounding peaks. However, by combining the GB and GR methods, the overall result is improved.

In the Resting video setting, the method Omit performed best, followed by GRGB, CHROM, and LGI, which performed equally well. In the Gym video setting, the GRGB method performed best, followed by POS. The two methods set themselves apart here and showed particular motion robustness. Together with the video setting rotation, Gym belongs to the video settings with the highest |ΔBPM| on average. In these two video settings, it can be seen that the GRGB method performed significantly better than the GR or GB methods individually, as shown in Table 2.

The light conditions in the ’Talk’ video setting are unique because they are natural light. In this case, our methods, which are based on simple fractions without any color channel normalization, showed modest performance. In this case, the CHROM method performed best, followed by POS. Again, it can be seen in Table 2, GRGB performed better than GR or GB individually, although not as clearly as in the video setting Gym or Talk.

The video setting Rotation had the highest |ΔBPM| on average and stood out with a particularly high level of head movements. Here, our proposed GRGB method also showed the best performance. This can be interpreted as the confirmation of the high level of motivation robustness. Method POS showed the second-best results in the video setting Rotation.

Overall, the GRGB method outperformed all of the other methods. Surprisingly, in the Gym video setting, the GRGB method was the best and showed high motion robustness. The GR method also performed well. In this study, the three best-performing methods are POS, LGI, and GRGB.

## 4. Discussion

It was also demonstrated that the GR method performed better than the green channel alone for every video setting. This can be interpreted as confirmation of the present study’s hypothesis. The sum of the two-color channel ratio applied in the GRGB method shows superior performance, although the single-ratio GB always performed worse than the GR. POS also leads to good results in our comparison, but the GRGB is simpler and less computationally complex. The time complexity of POS is $O(n^{2})$ while GRGB’s time complexity is O(1). POS requires multiple calculation steps, including temporal normalization for the chosen window length.

In the video recordings with a lot of motion and indoor lighting (e.g., Gym and Rotation), the GRGB method performed particularly well. Our GRGB method outperformed all the rPPG methods evaluated in this study. All of our proposed methods would produce the same rPPG signal at other RGB window lengths. Other methods, such as PCA or ICA, depend on the length of the signal used and eventually produce different results for different window lengths. In the existing literature, more than eight studies have introduced an rPPG method; at least seven of these studies reported results for indoor lighting conditions, which are worse than our simple GRGB method. These eight rPPG methods all have the same goals, and it is unclear under which conditions they perform best. More research is needed to understand which methods have better performance in specific environmental settings and under specific movement conditions. The LGI-PGGI dataset from Pilz et al. [14] is particularly suitable for investigation because it includes four different settings. For each video setting, there is a new ranking that, to the best of our knowledge, is not predictable. However, this dataset has limitations, as it consists mainly of Caucasian participants.

The literature almost unanimously ascertains that darker skin tends to make rPPG methods more error-prone [31,32]. However, there is no threshold value for melanin content at which the method’s performance decreases drastically [31]. Thus, the evaluated rPPG methods may perform differently for darker skin tones [33]. Therefore, more research should be conducted, ideally using a dataset that is representative of the general population in terms of skin tone. Furthermore, the participants tended to be young and healthy. Age could be a factor for algorithm potential bias [34]. It would also be interesting to compare participants who suffer from CVDs. Further investigation is needed to determine the possible influencing factors and to gain a deeper understanding of the relationship between the three color channels and the rPPG signal.

The GRGB method runs only one simple mathematical equation, making it an optimal rPPG approach for real-time applications, especially for artificial lighting conditions. The applied data set consists of six people, each with four video settings. A higher number of participants or more video material could help increase this study’s significance. This study showed that a simple ratio of red and green channels are an acceptable rPPG method for ideal environmental conditions (e.g., indoor lighting and minimal movements).

In addition to the environmental conditions during recording, the technical data of the videos are also relevant, such as resolution and frames per second. In the dataset used, the resolution is relatively low at 640×480, but it must be noted that the average values of the color channel in the ROIs per frame are used, as shown in Figure 1. Therefore, it can be assumed that higher resolutions can only slightly improve the result. In contrast, the frame rate probably has a stronger influence, as oximeters often work with higher frequencies. However, it must be taken into account that we have limited ourselves to a consumer-grade web camera. In summary, the video quality used is rather at the lower end of modern webcams, and the results could possibly be improved by a better web camera.

An increase in the complexity of rPPG methods can be observed in the literature. More specific methods are presented that are more suitable for specific scenarios or datasets. We greatly appreciate the work of the researchers who have investigated the current state of rPPG. The study discussed in this paper challenges the status quo and shows that much simpler methods can also produce sophisticated results.

We have provided the following recommendations for future research:1Examining the proposed GRGB method as a benchmark rPPG method.2Exploring the illumination changes. The lighting used in indoor settings is often not described well enough. To understand the rPPG method more deeply, it is essential to study the light that illuminates the face (e.g., spectrum or angle to the face).3Investigating the impact of ROI. Any movement of the subject can introduce artifacts into the extracted ROI, making it difficult to separate the signal from the noise. For video settings with little movement and constant light, we recommend focusing more on ROI selection.4Collecting videos from participants with different skin colors. We recommend testing all the examined algorithms on participants of different ethnicities and age ranges.

## 5. Conclusions

This study showed that using a ratio-based RGB method is a sufficient rPPG construction for HR estimation. Three ratio-based RGB methods were investigated: GR, GB, and GRGB. Both GR and GB provided comparable HR estimations to complex rPPG methods. Interestingly, the GRGB method achieved optimal performance compared to techniques with more sophisticated image processing algorithms capable of accurately detecting and tracking blood flow changes in real time. The proposed low-complexity and easy-to-implement method paves the future of rPPG technology, intending to make rPPG construction fast, accurate, robust, and accessible to a broader range of users.

## Figures and Tables

**Figure 1 bioengineering-10-00243-f001:**
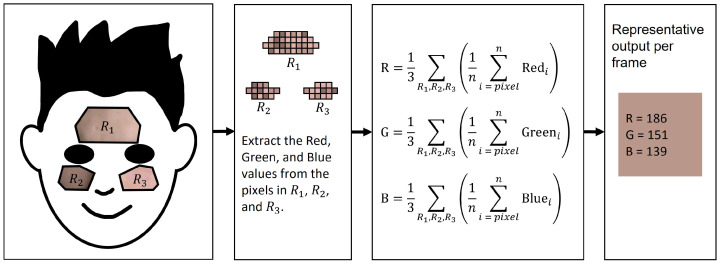
Graphical abstract of the pipeline from a frame to the R, G, and B values. The process repeats for each frame from the 1-min video resulting in the three time series: R(t), G(t), and B(t). *i* = pixel, *n* = number of pixels in ROI, $R_{1}$, $R_{2}$, $R_{3}$ = Regions of interests.

**Figure 2 bioengineering-10-00243-f002:**
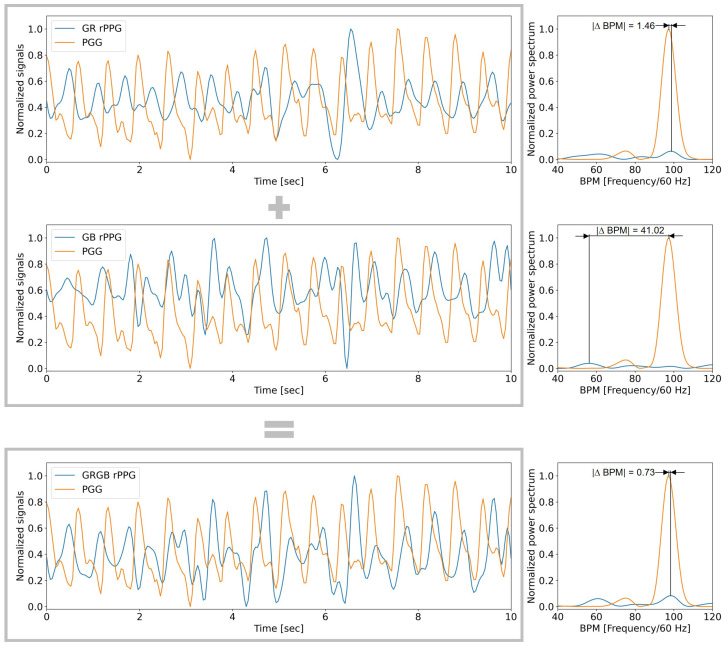
Representative normalized rPPG signals of the three proposed methods from a 10-second window in the Gym video setting of the participant, David, and the reference fingertip PPG signal. The combination of the GR and GB method results in the GRGB method.

**Table 1 bioengineering-10-00243-t001:** Summary of the introduced rPPG methods. Note: B = blue channels, G = green channel, R = red channel.

rPPG Method	Equation
Proposed Method I (GR)	$\mathrm{rPPG}\left(t\right)=\frac{G(t)}{R(t)}$
Proposed Method II (GB)	$\mathrm{rPPG}\left(t\right)=\frac{G(t)}{B(t)}$
Proposed Method III (GRGB)	$\mathrm{rPPG}\left(t\right)=\frac{G(t)}{R(t)}+\frac{G(t)}{B(t)}$

**Table 2 bioengineering-10-00243-t002:** Average |Δ BPM| for each video setting and rPPG method. A small |Δ BPM| means that the calculated BPM from the PPG and rPPG signals are in high agreement.

rPPG Method	Resting	Gym	Talk	Rotation	Average
Proposed Method I (GR)	1.99	11.64	8.52	18.33	10.12
Proposed Method II (GB)	1.99	29.74	10.27	22.44	16.11
Proposed Method III (GRGB)	1.91	6.84	7.98	12.90	7.41
CHROM [22]	1.91	14.81	3.68	18.84	9.81
LGI [14]	1.91	12.39	4.52	15.12	8.48
POS [11]	1.99	7.06	6.37	14.44	7.47
PBV [23]	2.95	22.73	13.08	26.16	16.23
PCA [21]	1.93	28.42	7.91	15.60	13.47
GREEN [13]	2.03	27.22	9.09	28.06	16.60
OMIT [24]	1.87	11.23	6.39	16.72	9.05
ICA [20]	8.10	26.00	14.16	22.38	17.66

## Data Availability

The dataset used is available at https://github.com/partofthestars/LGI-PPGI-DB (accessed on 1 January 2022 ). The applied rPPG methods are part of the pyVHR toolbox from https://github.com/phuselab/pyVHR (accessed on 1 January 2022).

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
