# Peer review of "GRGB rPPG: An Efficient Low-Complexity Remote Photoplethysmography-Based Algorithm for Heart Rate Estimation"

_bioengineering, 2023, doi:10.3390/bioengineering10020243_

Round 1

Reviewer 1 Report

Other than recommending an expanded study involving many different skin types and colors I have no suggestions that would improve this interesting paper. The authors may consider mentioning some pathophysiologic conditions that might be gleaned from capillary based pulse analysis such as capillary rarefaction as a result of diabetes and hypertension/salt sensitivity. 

Author Response

Author response: We thank the reviewer for the positive response and the valuable suggestion. 

Author action: We added the following sentences to address the raised point.

"The LGI-PGGI dataset from Pilz et al.12 is particularly suitable
for investigation because it includes four different settings. For each video setting, there is a new ranking that, to the best of
our knowledge, is not predictable. However, this dataset has limitations, as it consists mainly of Caucasian participants. The
literature almost unanimously ascertains that darker skin tends to make rPPG methods more error-prone.28, 29 However, there is
no threshold value for melanin content at which the method’s performance decreases drastically.28 Thus, the evaluated rPPG
methods may perform differently for darker skin tones.30 Therefore, more research should be conducted, ideally using a dataset
that is representative of the general population in terms of skin tone. Furthermore, the participants tended to be young and
healthy. Age could be a factor for algorithm potential bias.31 It would also be interesting to compare participants who suffer
from CVDs. Further investigation is needed to determine the possible influencing factors and to gain a deeper understanding of
the relationship between the three color channels and the rPPG signal.
An increase in the complexity of rPPG methods can be observed in the literature. More specific methods are presented that
are more suitable for specific scenarios or datasets. We greatly appreciate the work of the researchers who have investigated the
current state of rPPG. The study discussed in this paper challenges the status quo and shows that much simpler methods can
also produce sophisticated results."

Reviewer 2 Report

The authors have presented an important and efficient algorithm for the estimation of heart rate from remote photoplethysmography using a web camera. The paper is well written and the methodology and results have been presented in details. It is an important and interesting study; I have some comments as described below:

1. The results are promising. However, the sample is small as well as the duration of recording. Have the authors considered running a similar experiment on a different set of subjects and testing their hypothesis (or as a future study)?

2. It would be good to show the PPG signals for the four different settings. Furthermore, a figure showing the Power Spectrum plot would be informative.

3. It appears that the proposed method III (GRGB) achieved the best results and very comparable to the POS method. Can the authors please clarify the computational complexity of the GRGB versus the POS method; as well as elaborate on the benefits?  

Author Response

The authors have presented an important and efficient algorithm for the estimation of heart rate from remote photoplethysmography using a web camera. The paper is well written and the methodology and results have been presented in details. It is an important and interesting study; I have some comments as described below:

Author response: We thank the reviewer for the positive feedback and valuable suggestions.

Author action: None.

-------------------------------

1) The results are promising. However, the sample is small as well as the duration of recording. Have the authors considered running a similar experiment on a different set of subjects and testing their hypothesis (or as a future study)?

Author response: We thank the reviewer for the valuable suggestion.

Author action: We added the following sentences to clarify this point:

The GRGB method runs only one simple mathematical equation, making it an optimal rPPG approach for real-time
applications, especially for artificial lighting conditions. Further research on large datasets is needed to confirm the findings.
This study showed that a simple ratio of red and green channels are an acceptable rPPG method for ideal environmental
conditions (e.g., indoor lighting and minimal movements).
---------------------------------------

2. It would be good to show the PPG signals for the four different settings. Furthermore, a figure showing the Power Spectrum plot would be informative

Author response: We thank the reviewer for the valuable suggestion.

Author action: We added a new figure to address this point. Figure 2.

---------------------------------------

3. It appears that the proposed method III (GRGB) achieved the best results and very comparable to the POS method. Can the authors please clarify the computational complexity of the GRGB versus the POS method; as well as elaborate on the benefits?

Author response: We thank the reviewer for the valuable suggestion.

Author action: We added the following sentences to clarify this point:

It is important to note that the nine rPPG methods (OMIT22, GRGB, CHROM20, LGI12, PCA19, GR, GB, POS9, and GREEN11)
produced almost the same results. Other factors, such as the correct ROI or filter, could cause a significant difference in the
results. It was also demonstrated that the GR method performs better than the green channel alone for every video setting. This
can be interpreted as confirmation of the present study’s hypothesis. The sum of the two-color channel ratio applied in the
GRGB method shows superior performance, although the single-ratio GB always performed worse than the GR. POS also leads
to good results in our comparison, but the GRGB is simpler and less computationally complex. The time complexity of POS is
O(n2) while GRGB’s time complexity is O(1). POS requires multiple calculation steps, including temporal normalization for
the chosen window length.

Reviewer 3 Report

In this submission, the authors point out that rPPG is a promising contactless technology that uses videos of faces to extract health parameters, such as heart rate. Therefore, the authors focus on converting the RGB to rPPG based on RGB signals without applying complex mathematical models or machine learning algorithms. Finally, a series of experiments are conducted to show the effectiveness and efficiency of the proposed approach.

Generally, the proposed idea is of some novelty and the paper structure is organized well.

Therefore, I suggest a revise.

The authors should consider the following issues related to the manuscript quality.

1. The formulas and algorithms in the article should be written in a more standardized way.

2. Some related works can be added and, these works maybe help promote your work in wider communities, and make your work more informative.

“LSH-aware” multitype health data prediction with privacy preservation in edge environment."

"Time-aware missing healthcare data prediction based on ARIMA model"

3. Proofread the manuscript and correct the existing errors, such as the singular and plural forms of nouns, grammatical errors and the spelling mistakes.

4. The figures and tables can be refined to be more clearly and graceful by adjusting the layout of the figure body and legends.

Author Response

In this submission, the authors point out that rPPG is a promising contactless technology that uses videos of faces to extract health parameters, such as heart rate. Therefore, the authors focus on converting the RGB to rPPG based on RGB signals without applying complex mathematical models or machine learning algorithms. Finally, a series of experiments are conducted to show the effectiveness and efficiency of the proposed approach.

Generally, the proposed idea is of some novelty and the paper structure is organized well.

Author response: We thank the reviewer for the positive response and valuable suggestions.

Author action: None.

  1. The formulas and algorithms in the article should be written in a more standardized way.
    Author action: formulas are standardized per request.
  2. Some related works can be added and, these works maybe help promote your work in wider communities, and make your work more informative.

“LSH-aware” multitype health data prediction with privacy preservation in edge environment."

"Time-aware missing healthcare data prediction based on ARIMA model"
Author action: We cited the following papers to reach a wider communities

8. Rocque, G. B. & Rosenberg, A. R. Improving outcomes demands patient-centred interventions and equitable delivery. Nat.
Rev. Clin. Oncol. 19, 569–570 (2022).
9. Subbiah, V. The next generation of evidence-based medicine. Nat. Medicine 1–10 (2023).

  1. Proofread the manuscript and correct the existing errors, such as the singular and plural forms of nouns, grammatical errors and the spelling mistakes.
    Author action: the manuscript is proofread.
  2. The figures and tables can be refined to be more clearly and graceful by adjusting the layout of the figure body and legends.
    Author action: figures and tables are refined.